# A Deep Learning-Based Approach to Generating Comprehensive Building Façades for Low-Rise Housing

Da Wan [1,2], Runqi Zhao [1], Sheng Zhang [1,*], Hui Liu [1], Lian Guo [1], Pengbo Li [1] and Lei Ding [1]

1    School of Architecture, Tianjin Chengjian University, Tianjin 300380, China
2    Department of Architecture, Faculty of Environmental Engineering, The University of Kitakyushu, Kitakyushu 808-0135, Japan
*    Correspondence: zs_057@163.com

**Abstract:** In recent years, as machine learning has been widely studied in the field of architecture, scholars have demonstrated that computers can be used to learn the graphical features of building façade generation. However, existing deep learning in façade generation has yet to generate only a single façade, without comprehensive generation of five façades including the roof. Moreover, most of the existing literature has utilized the Pix2Pix algorithm for façade generation experiments, failing to attempt to replace the original generator in Pix2Pix with a different generator for experiments. This study addresses the above issues by collecting and filtering entries from the international Solar Decathlon (SD competition) to obtain a data set. Subsequently, a low-rise residential building façade generation model based on the Pix2Pix neural network was constructed for training and testing. At the same time, the original U-net generator in Pix2Pix was replaced with three different generators, U-net++, HRNet and AttU-net, for training and test results were obtained. The results were evaluated from both subjective and objective aspects and it was found that the AttU-net generative network showed the best comprehensive generation performance for such façades. HRNet is acceptable if there is a need for fast training and generation

**Keywords:** deep learning; generative adversarial network (GAN); façade generation; Pix2Pix; generator comparison

## 1. Introduction

With the rise of the third wave of artificial intelligence, machine learning represented by deep learning has been developed rapidly. The research and application of generative machine learning models have also made significant progress [1,2]. Some experts have pointed out that artificial intelligence (AI) has some unique creativity in conducting architectural design and exploration of architectural spaces [3]. These creativities may not be available to humans. This may suggest that the output of AI can be drawn upon for architectural design. Concurrently, some scholars argue that computers are unlikely to consciously design architectural works [4]. Some other experts and scholars have tried to apply machine learning methods to achieve the automatic generation of building plans [5–7] and façade layouts, to explore the possibility of using machine learning to realize applications in the field of architectural design [8].

Among generative models, the generative adversarial network (GAN) creatively proposed by Goodfellow in 2014, is composed of a generative network G and a discriminative network D, which can be used for image transformation [9]. Based on GAN, a conditional generative adversarial network (CGAN), was proposed, which introduces conditional variable y in both the generative network and the discriminative network, thus improving the quality of image generation [10]. Isola based on CGAN, proposed the Pix2Pix algorithm, in which the input image is fed into the discriminative network D together with the fake image generated by the generative network G for judgment, thereby obtaining the corresponding

output image [11]. In addition, some scholars have built the StyleGAN2 model, which can randomly generate reasonable building façade drawings through training [12]. Cycle-GAN can translate images without paired examples and apply them to color and mapping conversions [13]. Zheng Hao designed elements into coordinates and generates bedroom layouts by training an artificial neural network model, offering the possibility of generating multiple solutions [14]. In addition to this, Zheng Hao et al. have developed specific artificial neural networks as a means of learning and generating bedroom layouts through higher accuracy and faster in practice [15]. In a 3D generation, a voxel-based variational autoencoder approach has been trained and the generated models have been evaluated for application on the ModelNet benchmark, exploring the possibilities of deep learning in 3D modeling applications [16]. In practice, some scholars have used neural networks to train applications that can automatically generate building floor plans, an application that allows ordinary people to participate in architectural design [17]. "XKool" technology applies artificial intelligence technology to the whole cycle of real estate, enhancing design benefits, improving design efficiency, and tying together digital design full cycle management [18].

However, due to the limitations of the current GAN model's own neural network structure and training algorithm, most of the existing studies of façade generation are conducted for single façade generation studies. There are few comprehensive studies of the five building façades including the roof, which is not conducive to the designer's overall design. Meanwhile, the existing literature on building generation design still focuses on the elaboration of generation algorithms. But fails to adequately compare the generation results of different algorithms, which also makes designers feel overwhelmed when choosing the network. It also makes the choice of network overwhelming.

To address the above issues, this study establishes the GAN façade generation model, and explores the possibility of applying the GAN model in façade generation by selecting different generators to synthesize the training model. Additionally, it discusses the generation results of different generation networks. Findings from this study may provide an initial exploration of the performance disparity of different generators and provide an intuitive methodological basis for further design generation.

## 2. Materials and Methods

### 2.1. Research Framework

The entries of the SD competition were collected at the beginning. All the building façades were processed to obtain a comprehensive data set for training and testing. A series of façade generation models were built based on the Pix2Pix neural network, replacing the U-net generative network with U-net++, HRNet and AttU-net respectively. After training, the results of multiple sets of experiments were evaluated from both subjective and objective aspects (Figure 1).

### 2.2. Data Set Acquisition

This study uses the entries of the SD competition as a data set. The SD competition was initiated and hosted by the US Department of Energy in 2002 with universities worldwide as participants. Previous SD competitions were held in the US, Europe, China, Latin America, the Middle East, and Africa. The official websites recorded the most relevant documents from the recruitment to the official competition, where project manuals and technical drawings related to the entries can be accessed. After collection and screening, a total of 93 entries were obtained that could be processed as a data set (Table 1).

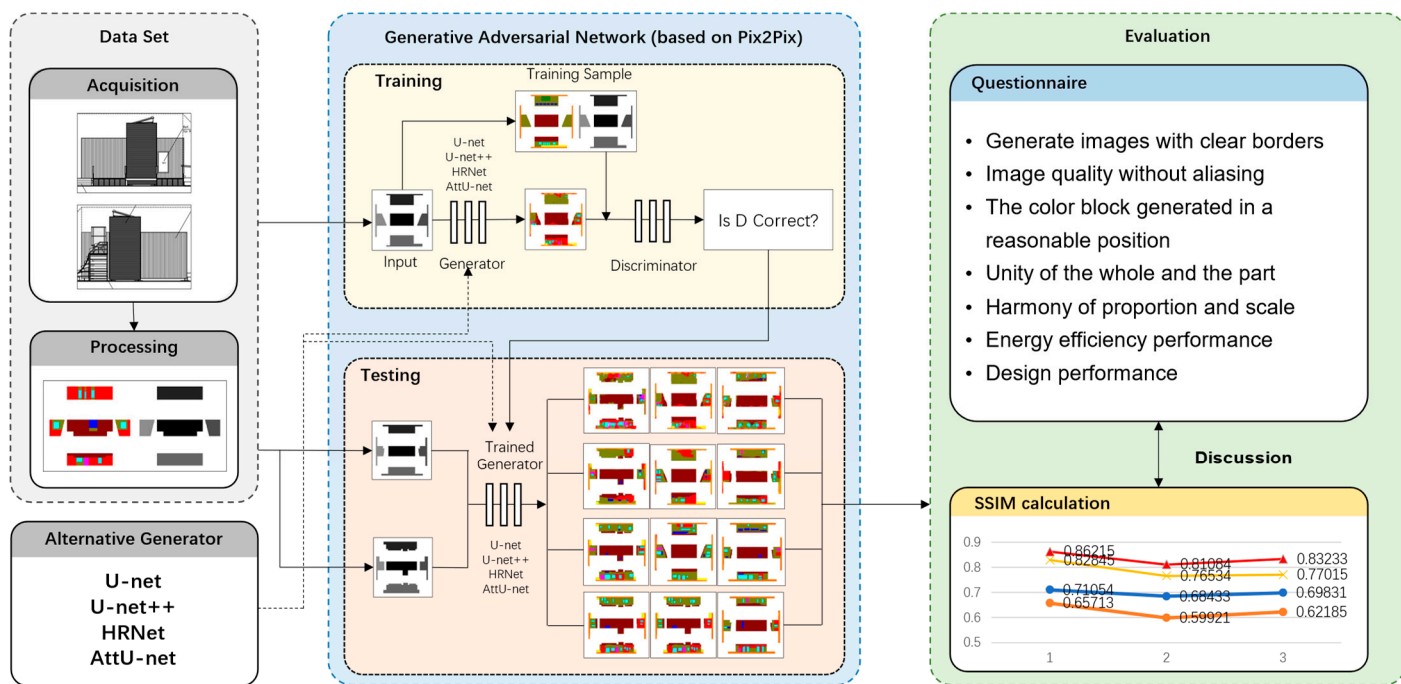

**Figure 1.** Research framework.

**Table 1.** Data collected and screened numbers.

| Competition | Location | Entries | Retained |
| --- | --- | --- | --- |
| SD2007 | Washington, DC, USA | 20 | 13 |
| SD2009 | Washington, DC, USA | 21 | 15 |
| SDE2010 | Madrid, Spain | 17 | 5 |
| SD2011 | Washington, DC, USA | 19 | 10 |
| SDE2012 | Madrid, Spain | 18 | 5 |
| SD2013 | Irvine, CA, USA | 20 | 15 |
| SD2015 | Irvine, CA, USA | 15 | 12 |
| SD2017 | Irvine, CA, USA | 11 | 9 |
| SDEM2018 | Dubai, UAE | 14 | 9 |
| Total | - | 155 | 93 |

*2.3. Data Set Processing*

Deep learning of low-rise residential facades focuses on learning the distribution of data within the image. For the Pix2Pix neural network to learn the components of the building facade such as roofs, doors, and windows, they need to be labeled and fed to the computer. The use of different colored blocks to represent the roofs, doors, windows, and other components of the building façade is the labeling process in deep learning.

This study first analyzed and filtered the SD entries, using a variety of color block maps as labels for the data, and used this to further process the data. An example of a standard format data set is shown in Figure 2. The image is 256 pixels high, 512 pixels wide, with a resolution of 300 dpi. The image on the left side is a labeled map of low-rise housing and the right side is a boundary map of the labeled one. The traditional black block cannot determine the orientation of the façade when processing the boundary map. Thus, the effect of orientation on the generation of the façade cannot be recognized when the generation carrying out.

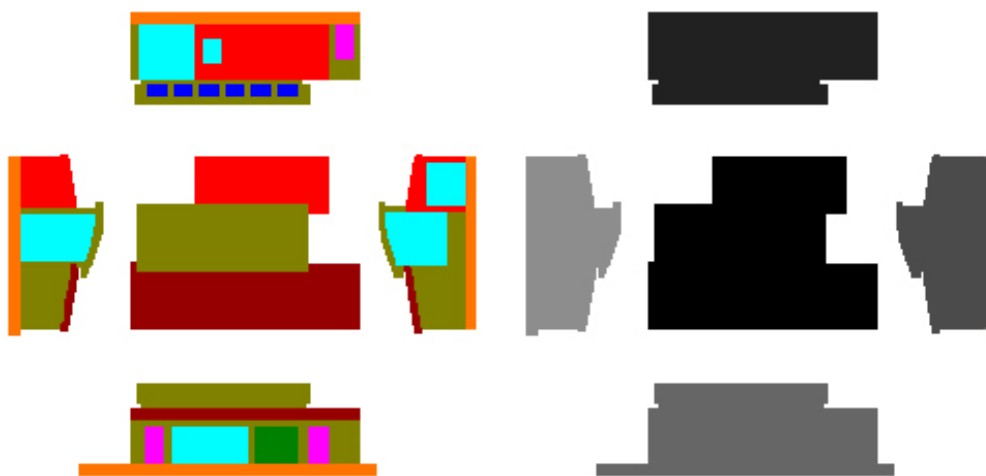

**Figure 2.** Standard data set.

Therefore, this study discarded the usual black block diagram when producing the data set and instead applied different transparency borders to the elevations of different orientations: 85% RGB (0, 0, 0) for north-facing elevations, 70% RGB (0, 0, 0) for east-facing elevations, 60% RGB (0, 0, 0) for south-facing elevations, 45% RGB (0, 0, 0) for west-facing elevations RGB (0, 0, 0) for the west-facing elevation, and 100% RGB (0, 0, 0) for the roof.

In the production of the labels, this study uses Photoshop software to label the built elements in the low-rise residential façade, each component having its corresponding RGB color, with the detailed color settings shown in Table 2.

**Table 2.** Component color comparison table.

| Name | Color (R, G, B) | Name | Color (R, G, B) | Name | Color (R, G, B) |
|---|---|---|---|---|---|
| Photovoltaic panels | (128, 0, 0) | Door | (255, 0, 255) | Windows | (0, 255, 255) |
| Plain walls | (128, 128, 0) | Wooden walls | (255, 0, 0) | High windows | (0, 0, 255) |
| Greening | (0, 128, 0) | Railings | (255, 255, 0) | Steps | (255, 115, 0) |

In RGB colors, there are 256 levels of brightness each, expressed numerically from 0 to 255, and these can usually be divided into three stages. The first is from 0 to 100; the second from 101 to 200; and the third from 201 to 255. In the selection of values for R, G and B, each color is guaranteed to have at least two stages of values.

### 2.4. Generating Network Selection

### 2.4.1. U-Net Generation Network

The U-net network structure is based on the Encoder-Decoder convolution and deconvolution operations. As well, the Encoder-Decoder based model is modified by adding a skip-connection, so that the left and right sides of the structure are directly connected, and layer i is directly connected to layer n − i, thus mapping the encoder output. The aim of the convolution process is to convolute the image to the right. The purpose of the convolution process is to extract the image features and compress the image, while the purpose of the deconvolution process is to up-sample the image size to achieve the original resolution.

The U-net generative network used in this study is set up with 5 layers of convolution and 5 layers of deconvolution, and the network structure of the U-net is shown in Figure 3.

The padding of all five layers is set to 1. The deconvolution layers are also up-sampled using a 3 × 3 convolution kernel and a 2 × 2 deconvolution, and the padding of all five deconvolution layers is set to 1. The input of each deconvolution layer in the generative network structure includes both the output of the previous layer and the output of the corresponding convolution layer. The input to each layer of the generative network structure includes the output of the previous layer and the corresponding convolution layer, so that the generated image retains as much information as possible from the original image.

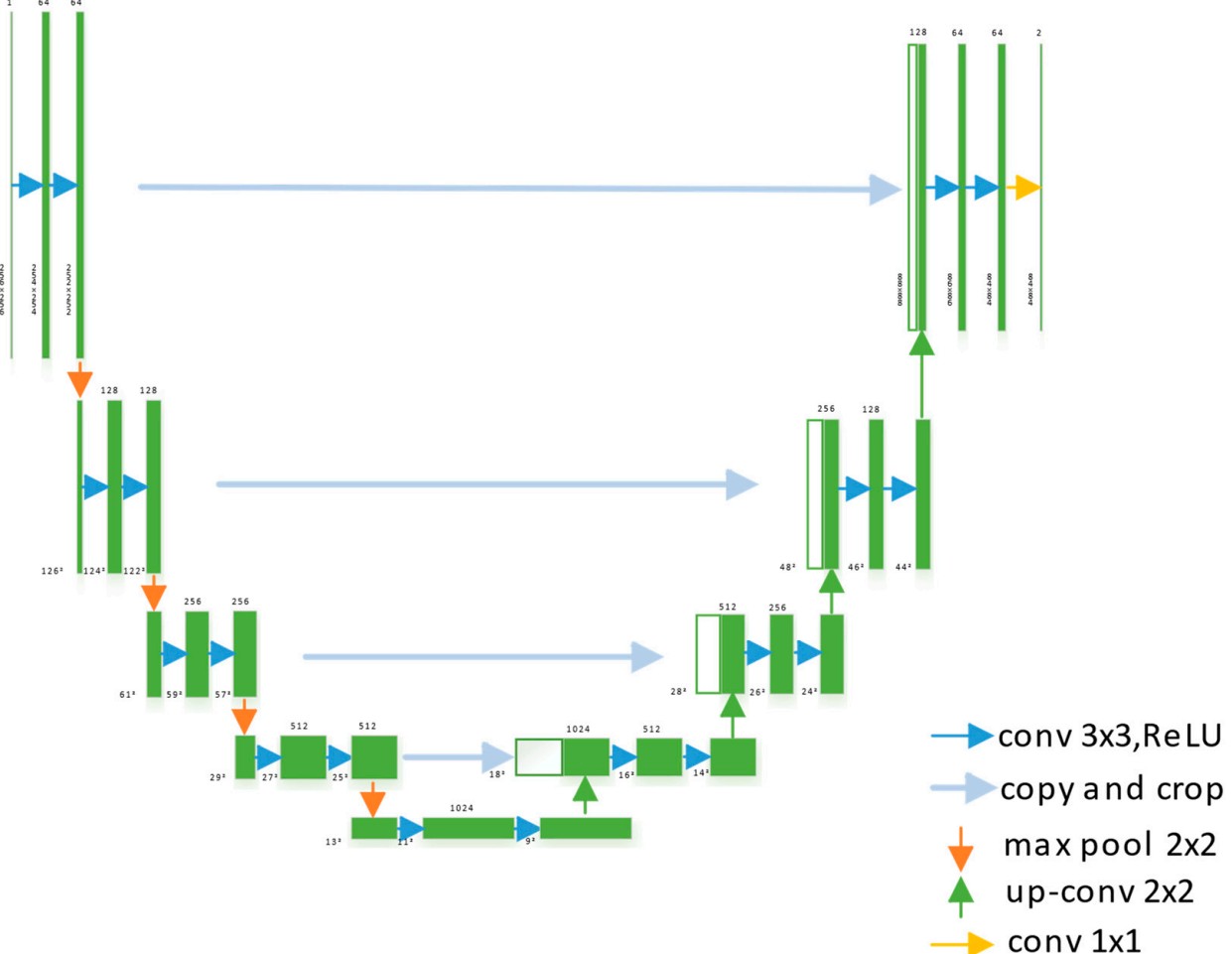

**Figure 3.** U-net generation network structure (based on 'U-net: Convolutional Networks for Biomedical Image Segmentation' [19]).

### 2.4.2. U-Net++ Generation Network

The U-net model is based on an encoder-decoder structure, while the U-net++ is based on the U-net model, combined with DenseNet and deep supervision principles. Its main network structure is shown in Figure 4 [20].

The U-Net++ network with its nested structure and dense jump paths has a great advantage in extracting feature maps from multi-level convolutional paths. The biggest difference between U-net++ and U-net is the redesigned jump paths in U-net++. Take node $X^{0,4}$ as an example, in the U-net model structure, node $X^{0,4}$ simply constructs a jump connection with node $X^{0,0}$ a jump connection. In U-net++, node $X^{0,4}$ connects the outputs of the four convolution units $X^{0,0}$, $X^{0,1}$, $X^{0,2}$ and $X^{0,3}$, which are at the same layer. This structure of the U-net++ network enables the semantic level of the feature map within the encoder to be closer to the semantic level of the corresponding decoder part.

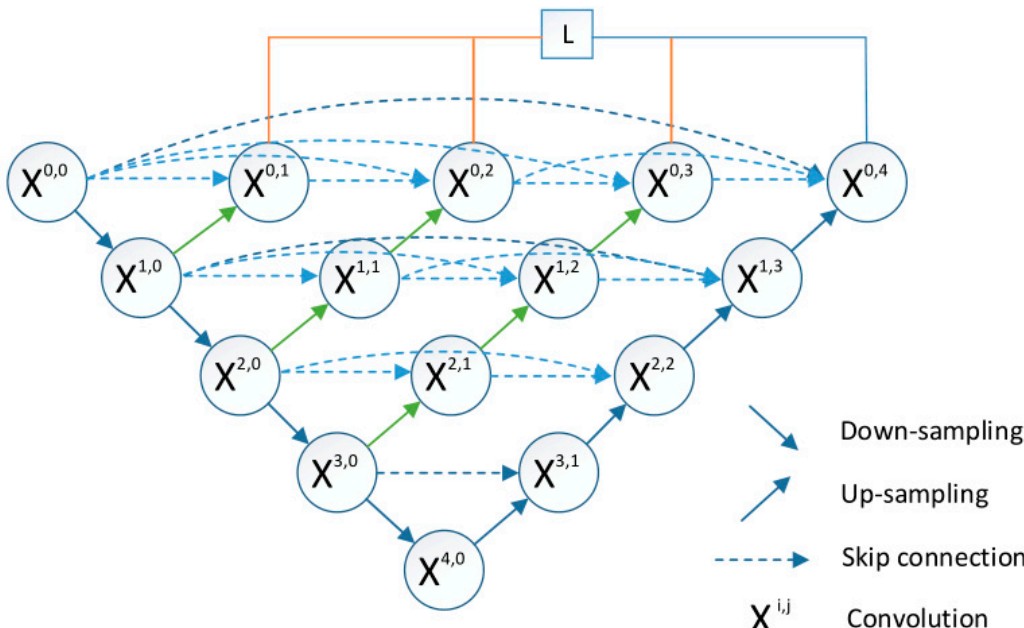

**Figure 4.** U-net++ generation network structure. (based on 'Unet++: A Nested U-net Architecture for Medical Image Segmentation' [20]).

### 2.4.3. HRNet Generation Network

HRNet (high-resolution Net) was proposed in 2019 and has achieved good results in keypoint detection, pose estimation, and multi-person pose estimation [21]. To verify the effectiveness of the HRNet network for building façade extraction, this study replaces the U-net generative network in the Pix2Pix architecture with the HRNet generative network for experiments.

The HRNet neural network is a parallel structure that acts as an image feature skeleton extraction network. It uses $3 \times 3$ convolution for deeper down sampling while maintaining 4 times downsampling resolution to expand the perceptual field and extract deeper information about the image with a minimum resolution of 1/32 of the original image. Use the BasicBlock module in the forward propagation of the feature map of the same resolution and set the step size to 1. At the same time, the feature maps between different resolutions maintain information interaction, with multiple convolutions with a step size of 2 from high resolution to low resolution. As well, bilinear interpolation up sampling from low resolution to high resolution. Finally, information from different resolutions is received at each layer and stitched together in the channel dimension to complete the information fusion. Each façade in the data set of this study has obvious block features, and its contextual semantic features are obvious. Special attention needs to be paid to the information on the boundaries when detecting the façade, and HRNet may make a great achievement in this regard.

### 2.4.4. AttU-Net Generation Network

In a normal convolutional network, the value of the target pixel is only calculated with reference to itself and the surrounding pixels. This means that convolution can only use local information to compute the target pixel, which may introduce some bias because the global information is not visible. In this study, the self-attention mechanism is introduced into the part of the down sampling of the U-net connected with the corresponding up sampling layer, and the hopping connection structure is preserved. This allows the network to fully mine the global information and extract some details in the façade more accurately. The addition of the Attention gate to the U-net adds very little additional computation yet brings significant improvements in model sensitivity and accuracy, achieving a global reference for each pixel-level prediction [22].

The AttU-net network consists of an Encoder, a Decoder, and the Attention gate. A self-attentive module that passes through the encoding convolution module, extracts the bottom-level features, and then feeds a downsampling block to reduce the spatial size and obtain high-level features. The number of channels is doubled with each down-sampling block, and the end of the down-sampling is fed to the Attention module, which aggregates the global information and produces the output of the encoder.

*2.5. Evaluation*

The evaluation and discussion of the generated results are also crucial when using the generative model to automate the design of low-rise residential façades, using a combination of subjective and objective evaluation methods in the evaluation

2.5.1. Subjective Evaluation

This study developed a façade results questionnaire for professionals based on the quality of the generated images, traditional architectural design requirements, as well as the subjective opinions of professional academics. The scoring table first subjectively evaluates the boundary condition and color block quality of the generated images, then evaluates whether the position, size, and proportion of the color blocks representing the façade components meet the requirements from the traditional architectural design perspective, as well as the architectural professionals to evaluate the performance of the generated results and the façade design effect from the subjective aspect. The façade results in evaluation table are shown in Table 3.

**Table 3.** Questionnaire scoring.

| Marking Content | Score |
|---|---|
| Generate images with clear borders | 0~5 |
| Image quality without aliasing | 0~5 |
| The color block generated in a reasonable position | 0~5 |
| Unity of the whole and the part | 0~5 |
| Harmony of proportion and scale | 0~5 |
| Energy efficiency performance | 0~5 |
| Design performance | 0~5 |

2.5.2. Structural Similarity Evaluation (SSIM)

In addition to relying on some subjective façade evaluation guidelines for the evaluation of low-rise residential façade generation results, this study also introduces objective evaluation criteria. After comparing a number of image quality evaluation criteria, the SSIM structural similarity evaluation metric was chosen as a better match to the generative design approach proposed in this study.

The SSIM structural similarity metric is based on the human visual system and can be used to measure the distortion of an image as well as the similarity between two images [23]. SSIM focuses on the consistency of the image in terms of brightness and color, while considering high-frequency information such as image edges and details.

## 3. Results

*3.1. Data Screening*

The purpose of data filtering is to filter and clean the data collected, remove large discrepancies and incorrect data, and improve data consistency.

A total of 166 entries from 12 years of the SD competition were collected for this study. After collecting the competition entries, the entries were screened to select those suitable for façade generation for training and testing. We followed two selection bases: (1) By the number of building stories. The majority of the competition's requirements for the number of stores are for single-story buildings. As well, multi-story buildings do not occupy a high proportion of the data set, so we only retain single-story entries. (2) By building form. In

the past years, according to the layout of the building form, it can be roughly divided into four categories: cluster, independent block, irregular and heterogeneous. When selecting the works, we remove the cluster, irregular and heterogeneous categories, and only keep the relatively regular works, so as to facilitate the computer to learn and generate.

Through screening, a total of 93 valid and usable works were finally obtained. Out of the 93 entries, three samples were randomly selected as the test set and the remaining 90 samples were used as the training set for training. Ninety samples is a small sample size for deep learning, so this study augmented the data by rotating the geometric transformation by 90°, 180°, and 270°. The number of training sets after augmentation reached 360.

### 3.2. Experimental Configuration and Parameter Settings

3.2.1. Experimental Environment Configuration

The computer environment configuration settings for the model training experiments are shown in Table 4.

**Table 4.** Experimental environment configuration parameters.

| Item | Configuration | Item | Configuration |
| --- | --- | --- | --- |
| Operating systems | Windows10 | Compilers | PyCharm |
| CPU | AMD2600X | CUDA | CUDA10.1 |
| GPU | RTX2080Ti | CuDNN | CuDNN7.6.5 |
| Development Languages | Python3.7 | Deep Learning Framework | Pytorch1.7.1 |

3.2.2. Parameter Setting

1.　Learning rate

The empirical value of 0.0002 is often used for the learning rate, and the learning rate was set to 0.00002, 0.0002, and 0.002 for the experiments. It was found that when the learning rate was 0.00002, the gradient decreased slowly. When the learning rate was 0.002, it converged too fast but crossed the optimal value. When the learning rate was 0.0002, it could converge to the lowest value in a suitable time. Therefore, the experimental learning rate settings for this study were all 0.0002.

2.　Number of iterations Epoch

An epoch is when all the data is put into the network for a single forward calculation and backpropagation. However, the training of a model often requires many iterations to reach a state of convergence. In this study, the model was under-fitted when the number of iterations was 300, moderately fit when the number of iterations was 400, and over-fitted when the number of iterations was 500. Therefore, the number of iterations for all experiments in this study was set to 400.

### 3.3. Generating Results by Using the U-Net Network

Table 5 shows the results generated using the U-net network with a learning rate set to 0.0002 and a number of iterations of 400. The input to the model is the building façade boundary image, the training Ground Truth is the building façade layout color block image, and the output is the output building façade layout color block image, taking a total of 2.5 h.

### 3.4. Generating Results by Using the U-Net++ Network

Table 6 shows the generated results using the U-net++ network, with the learning rate set at 0.0002 and the number of iterations at 400. The input to the model is the building façade boundary image, the trained Ground Truth is the building façade layout color block image, and the output is the output building façade layout color block image, taking a total of 3.5 h.

**Table 5.** Generating results by using the U-net network.

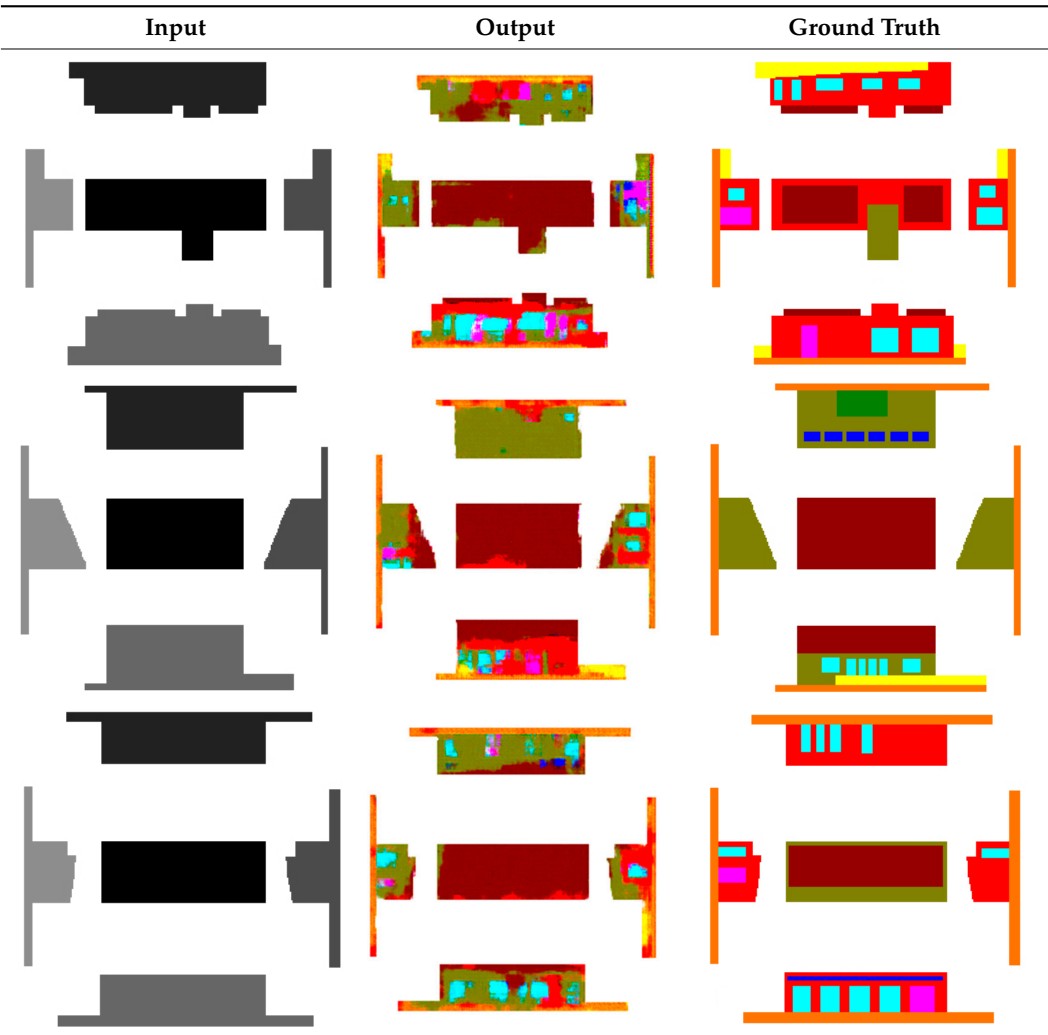

**Table 6.** Generating results by using the U-net++ network.

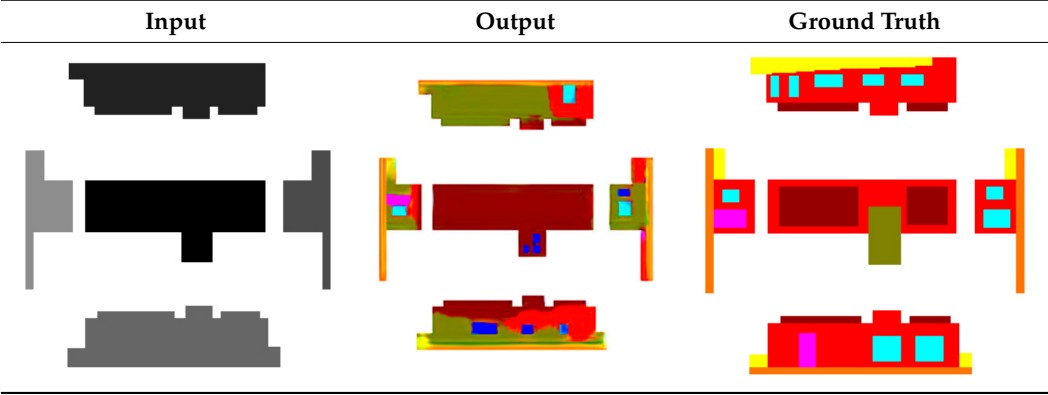

**Table 6.** *Cont.*

| Input | Output | Ground Truth |
|-------|--------|--------------|

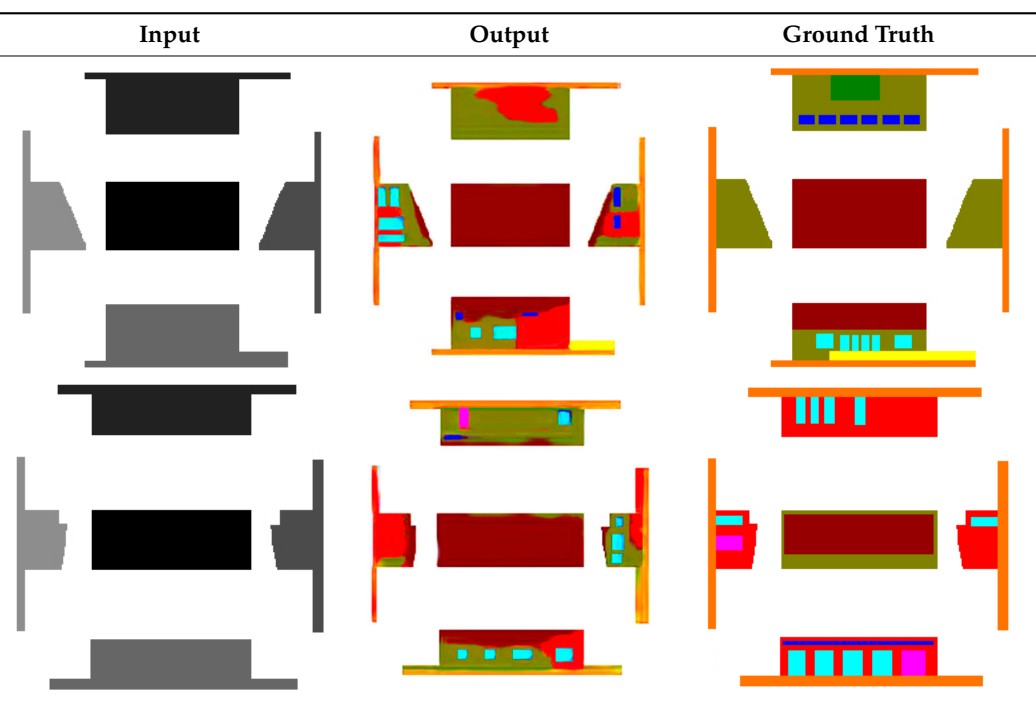

### 3.5. Generating Results by Using the HRNet Network

Table 7 shows the generated results using the HRNet network, with the learning rate set at 0.0002 and the number of iterations at 400. The input to the model is the building façade boundary image, the trained Ground Truth is the building façade layout color block image, and the output is the output building façade layout color block image, taking a total of 3 h.

**Table 7.** Generating results by using the HRNet network.

| Input | Output | Ground Truth |
|-------|--------|--------------|

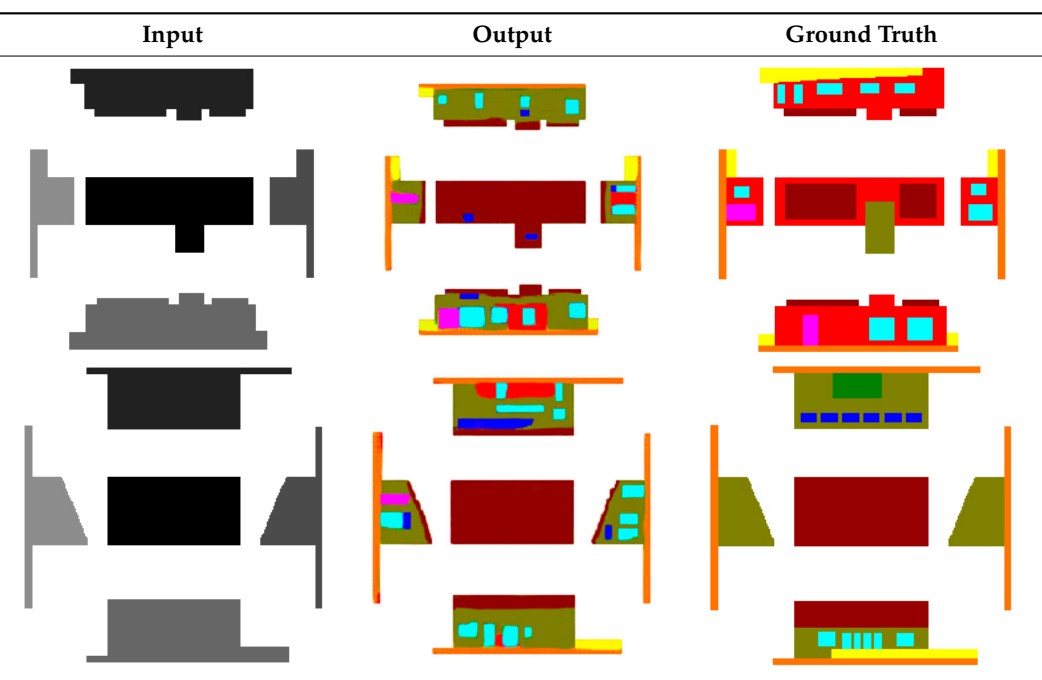

**Table 7.** *Cont.*

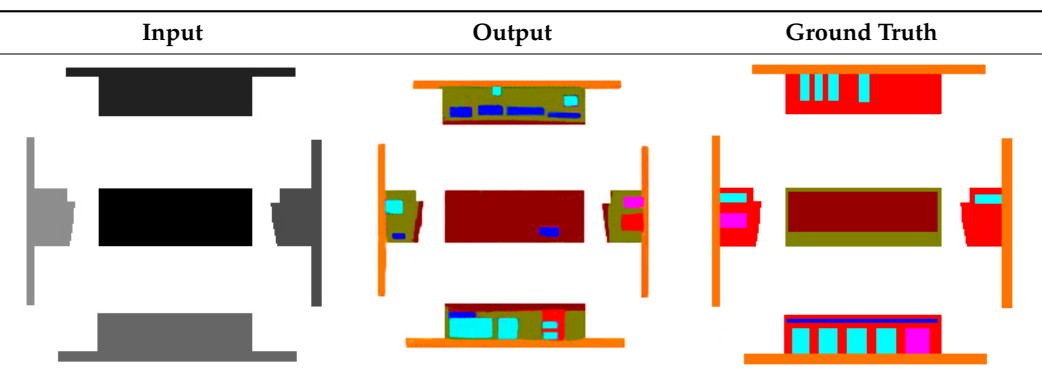

### 3.6. Generating Results by Using the AttU-Net Network

Table 8 shows the generated results using the AttU-net network with a learning rate set to 0.0002 and 400 iterations. The input to the model is the building façade boundary image, the trained Ground Truth is the building façade layout color block image, and the output is the output building façade layout color block image, taking a total of 4 h.

**Table 8.** Generating results by using the AttU-net network.

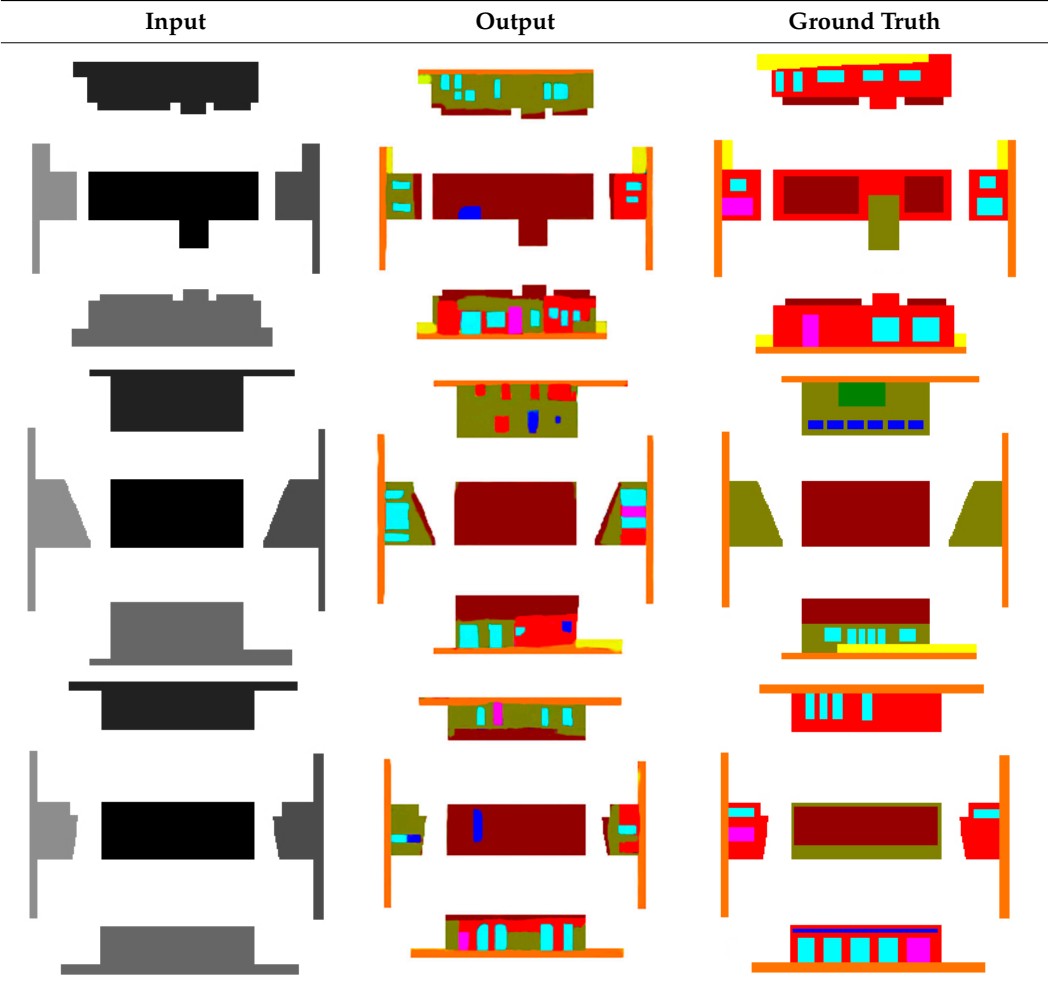

## 4. Discussion

### 4.1. Subjective Evaluation

The questionnaire was distributed to architecture professionals in this study and 36 copies returned. The feedback scores are shown in Table 9. The results of the feedback reveal that HRNet and AttU-net scored higher in terms of generating image boundaries. u-net's results suffer from blurred façade boundaries and U-net++ is slightly improved, but still slightly inadequate. In terms of miscellaneous colors in the functional color block diagram, AttU-net performs better and U-net scores lower.

From a traditional architectural design point of view, AttU-net and HRNet perform better in terms of the placement of functional color block generation, the unity of the whole and the parts, and the coordination of proportion and scale. U-net and U-net++ score lower. Some architectural professionals believe that the generated results of AttU-net may perform better in terms of performance. U-net, on the other hand, is likely to perform less well in terms of performance. In terms of façade design results, professionals scoring results indicated that the AttU-net generator and the HRNet generator had acceptable façade design results.

In terms of active energy efficiency measures, none of the generators generated green roofs and vertical greening, which may be related to the low number of cases with green roofs and vertical greening in the training set. However, both the AttU-net generator and the HRNet generator performed better in generating photovoltaic panels, which had a regular shape and were generated in a reasonable position.

**Table 9.** Scoring results of the questionnaire.

| Questionnaire Item | U-Net | U-Net++ | HRNet | AttU-Net |
|---|---|---|---|---|
| Generate images with clear borders | 3 | 3.4 | 4 | 4.1 |
| Image quality without aliasing | 2.5 | 3.1 | 3.7 | 4 |
| The color block generated in a reasonable position | 2.7 | 3.2 | 3.8 | 3.9 |
| Unity of the whole and the part | 2..7 | 3.2 | 3.8 | 4.2 |
| Harmony of proportion and scale | 3 | 3.3 | 4.1 | 4.2 |
| Energy efficiency performance | 2.6 | 3.1 | 3.7 | 4.1 |
| Design performance | 2.7 | 3.1 | 3.8 | 4.1 |
| Average score | 2..7 | 3..2 | 3.8 | 4.1 |

### 4.2. Objective Evaluation

The SSIM structural similarity was used for the objective evaluation. However, the calculation of SSIM values by comparing the generated results with which data still needs to be explored. In this study, two evaluation methods are proposed: the first is the calculation of SSIM values from the generated results to the ground truth. The second is the calculation of SSIM values from the generated results to the input scheme (color block labeled maps).

This study investigates the effectiveness of different generators in Pix2Pix to produce good or bad results. In the first method, the results generated differ significantly from the ground truth in terms of color and brightness, and it is not possible to judge the effectiveness of the generators. So, the second method is chosen for the calculation of SSIM values.

The experiments were conducted by random sampling and three samples were randomly selected from 93 data sets as the test set. After the training was completed, the three test sets were input into the test program for testing, and the generated results were compared with the input solution to calculate their SSIM values. The output SSIM values are shown in Table 10, and their average results were calculated for the SSIM values of each generated result.

**Table 10.** SSIM values and mean value calculation.

| Generators Category | U-Net | U-Net++ | HRNet | AttU-Net |
|---|---|---|---|---|
| Sample 1 SSIM values | 0.65713 | 0.71054 | 0.82845 | 0.86215 |
| Sample 2 SSIM values | 0.59921 | 0.68433 | 0.76534 | 0.81084 |
| Sample 3 SSIM values | 0.62185 | 0.69831 | 0.77015 | 0.83233 |
| SSIM average | 0.626 | 0.697 | 0.788 | 0.835 |

The value of SSIM ranges from 0 to 1. When SSIM = 1, it means that the two images are identical. When the value of SSIM is smaller, it means that the difference between the generated image and the target image is greater. The SSIM values for the three samples in the four sets of networks are represented in Figure 5

The AttU-net scored the highest at over 0.8, which is consistent with the subjective evaluation on sharpness and color purity. The HRNet was slightly lower than 0.8 and reached 94% of the AttU-net score. If there is a need for fast training and solution generation, HRNet might be capable of performing.

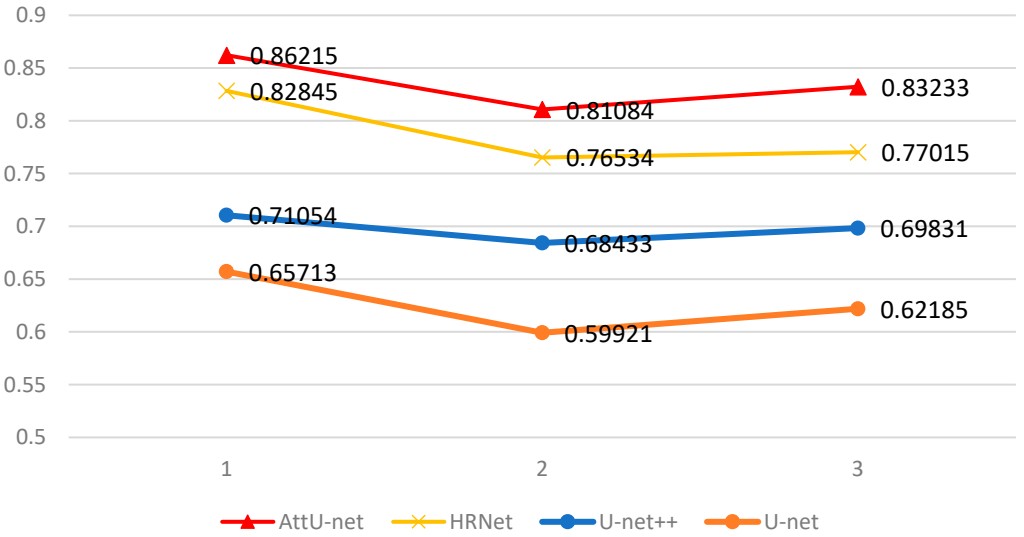

**Figure 5.** SSIM values comparison.

## 5. Conclusions

In this study, a building façade generation method based on the generative adversarial network is proposed. Taking the entries of the SD competition as an example, the Pix2Pix algorithm is used to build a generative model of a low-rise residential building façade and automatically obtain the layout results of the low-rise residential building façade. The U-net generating network in Pix2Pix was also replaced with U-net++ generating network, HRNet generating network, and AttU-net generating network respectively. The results of the four-generation networks were surveyed by questionnaire and their SSIM values were calculated separately.

1.  The subjective evaluation showed that the AttU-net generator and the HRNet generator had acceptable façade design results in terms of façade design results.
2.  The generated results show a certain degree of energy efficiency, especially the reasonable shape and position of the photovoltaic panel.
3.  The average structural similarity between the results of the AttU-net generation network and the target color block diagram was greater than 0.8. Indicates that the replacement of the U-net generation network of Pix2Pix with the AttU-net generation network in this study can generate a more reasonable comprehensive building façade layout.

4. Compared with traditional parametric design, the method used in this study is able to use deep learning to discover the patterns in the façade layout without human intervention. It can generate façade layouts efficiently.

5. AttU-net has the best comprehensive performance. Considering that approximately 25% of training time can be saved, HRNet is another acceptable choice in scenarios where there is a need for fast training and generation. The subjective scores of its generated results are 7% lower than AttU-net and 6% lower in SSIM value.

The approach proposed in this study is still a preliminary application of generative adversarial networks in the automatic generation of building façade layouts, and still has some limitations in terms of performance. The stability of the model training and the clarity of the generated results still need to be improved. Further expansion of the training data set is necessary to overcome the issues of insufficient training and the long training time of the output model. In the future, there is still a need to study the overall generation of floor plans and elevations, as well as, the integration of active and passive strategies to build a more comprehensive model of energy-efficient housing.

**Author Contributions:** Conceptualization, D.W. and R.Z.; methodology, R.Z. and H.L.; software, R.Z.; validation, R.Z. and L.G.; formal analysis, D.W. and R.Z.; investigation, R.Z. and L.D.; resources, R.Z. and H.L.; data curation, D.W., R.Z. and P.L.; writing—original draft preparation, D.W. and R.Z.; writing—review and editing, D.W. and R.Z.; visualization, R.Z.; supervision, D.W. and S.Z.; project administration, S.Z.; funding acquisition, H.L. and P.L. All authors have read and agreed to the published version of the manuscript.

**Funding:** This work was supported by the National Key Research and Development Program of China (grant number 2019YFD1100402), the Key Technology Research and Development Program of Shandong (grant number 2019GSF110004) and the Building Physical Environment Simulation (grant number 2021H0_0076).

**Institutional Review Board Statement:** No applicable.

**Informed Consent Statement:** No applicable.

**Data Availability Statement:** Data available on request due to restrictions, e.g., privacy or ethics.

**Acknowledgments:** We thank Jia Liu and Ivy Wang (Wan An Secondary School, Shanxi, China) for their valuable comments on this article. As well as the DAMlab (Digital Architecture and Manufacture Laboratory) Tianjin Sector in Tianjin Chengjian University for providing highly qualified equipment support during this study.

**Conflicts of Interest:** The authors declare no conflict of interest.

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
