# Peer review of "A Deep Learning-Based Approach to Generating Comprehensive Building Façades for Low-Rise Housing"

_sustainability, doi:10.3390/su15031816_

Round 1

Reviewer 1 Report

1. The picture can be clearer, as shown in Figure 1 and Figure 2

2. References need to be re-edited, such as references 1 and 2, which are problematic

3. How to compare subjective and objective evaluation in the conclusion of the article.

Author Response

Dear reviewer,
Thank you for your constructive comments on our manuscript entitled "A Deep Learning-Based Approach to Generating Comprehensive Building Façades for Low-Rise Housing" (ID:sustainability-2152073). Those comments are very helpful for revising and improving our paper, as well as the important guiding significance to further research. We have studied the comments carefully and made corrections which we hope meet with approval. The main corrections are in the manuscript and the responds to the your comments are as follows.

Redraw the research framework. 
Add questions regarding both design and energy efficiency performance to the questionnaire for subjective evaluation.
Expand the application scenarios of different algorithms by comparing subjective and objective evaluations.
Add the bullet points to highlight the contributions, as well as enhance the organization of the conclusions.
Slightly modify the manuscript structure to make it more compatible with the reading habits of most readers.
Address more elaboration and labeling on the figures and tables.
Unify some non-standard nouns and improve English expressions throughout.

We appreciate for your valuable comments, and hope that the correction will meet with your approval.
Looking forward to hearing from you.

Yours sincerely,
Da Wan & Runqi Zhao

Reviewer 2 Report

The paper concerns the use of a deep learning-based approach to generate comprehensive building façades for low-rise buildings. In recent years, machine learning has been used to understand the graphical generation of building façades. However, it was limited to only one façade, making it impossible to evaluate the design of the building as a whole. This paper attempts to comprehensively generate the elevation on four sides and the roof.

Most of the existing literature used the Pix2Pix algorithm for elevation generation experiments, without trying to replace the original generator with another generator. The authors collected and filtered data from the international Solar Decathlon competition organized by the US Department of Energy since 2002, in which universities from around the world participated. Through selection, a total of 93 valid and usable works were finally obtained. To increase the deep learning sample size, the data was augmented by rotating the geometric transformation by 90°, 180° and 270 °. The number of training sets after the increase reached 360.

The study first analyzed and filtered the data using different color block maps as labels for the data and used this to further process the data. Each element of the residential low-rise façade was given a previously defined RGB color. Subsequently, a low-rise residential building façade generation model based on the Pix2Pix neural network was constructed for training and testing.

The original U-net generator in Pix2Pix has been replaced with three different generators, U-net++, HRNet, and AttU-net, for training and test results.

The authors examined the possibility of using the GAN model to generate facades by selecting different generators for the synthesis of the training model and discussed the results of generating different generation networks. The results after the experiments were assessed both subjectively and objectively, and the AttU-net generative network was found to be the best performer in performing complex generation of such façades.

To sum up, it should be stated that despite the fact that generative machine learning models have made great progress, they are still insufficient in analyzing architectural issues. An example is this paper, the results of which are unsatisfactory and require significant improvement. However, considering this study as a preliminary application of adversarial generative networks in the automatic generation of façade building systems, it should be evaluated positively.

Author Response

(The authors gave the same response as above.)

Reviewer 3 Report

Dear Authors,

It was a pleasure to review your work. I have a few suggestions that you could consider:

1) The introduction and/or discussion could be enriched with a view on the perspective in which computers replace architects. Is creativity still a unique feature of human designers? Are human architects replaceable? Such considerations and other issues related to the CAD technology were the theme of recently published spacial issue by MDPI Buildings journal:

https://www.mdpi.com/journal/buildings/special_issues/Artl_Engineering

as well as in Sustainability:

https://www.mdpi.com/journal/sustainability/special_issues/View_of_Sustainability

2) Paper structure: I am unsure whether presenting part of the results at the end of the introduction is a good practice. I realize the intention was to briefly introduce the reader to the research outcomes, which could be made. But at this stage, it is not very convincing to the reader. For example, when you state in Figure 1 that AttU-net had the best results, it is not very convincing at this moment, and it makes the impression that you actually intended to achieve such a result from the beginning of the research. Maybe the right place for Figure 1 would be in the Methods section. Please rethink the organization of the content. 

3) With regard to the "2.4.1. Subjective Evaluation Methods" section: it is indispensable to inform the reader how the scoring was performed. Were there some experts asked to provide this scoring? How many were they? Were they architects or other professionals? etc. 

4) The discussion chapter should show your results on the background of the existing state of knowledge. One of the ideas that could be included is the question of aesthetics and spatial order. Are the computer-generated facades as aesthetic as those designed by humans? Maybe they are even more efficient in achieving the spatial order?

Author Response

(The authors gave the same response as above.)
